# Evaluation of Optimized Lumbar Oblique X-Ray Angles with Positioning Assistance for Enhanced Imaging Quality: A Pilot Study in an Asian Cohort

**DOI:** 10.3390/jfmk10010023

**Published:** 2025-01-05

**Authors:** Yu-Li Wang, Hsin-Yueeh Su, Chao-Min Cheng, Kuei-Chen Lee

**Affiliations:** 1Department of Surgery, Hualien Armed Force General Hospital, Hualien 971, Taiwan; yuli0329@yahoo.com.tw; 2International Intercollegiate Ph.D. Program, National Tsing Hua University, Hsinchu 300, Taiwan; 3Department of Radiology, Hualien Armed Force General Hospital, Hualien 971, Taiwan; levis512@gmail.com; 4Institute of Biomedical Engineering, National Tsing Hua University, Hsinchu 300, Taiwan; 5Department of Physical Medicine and Rehabilitation, Tri-Service General Hospital, National Defense Medical Center, Taipei 114, Taiwan

**Keywords:** oblique examination, positioning auxiliary device, 3D manufacture, spondylolisthesis, image quality

## Abstract

**Objective:** Pars fractures are a common cause of lower back pain, especially among young individuals. Although computed tomography (CT) and magnetic resonance imaging (MRI) scanning are commonly used in developed regions, traditional radiography remains the main diagnostic method in many developing countries. This study assessed whether the standard radiographic angles suggested in textbooks are optimal for an Asian population since Asian groups have lower lumbar lordosis. This study found a 35° angle to be the most effective angle for lumbar oblique X-ray imaging. Additionally, the potential for a customized positioning auxiliary device was examined to improve image quality and reduce patient discomfort in cost-sensitive healthcare settings like Taiwan’s single-payer system. **Methods:** A total of 100 participants underwent lumbar oblique radiography using a specially designed footboard with angle markings. Radiologists evaluated 600 images based on waist-to-hip ratio (WHR) and body mass index to identify the optimal angulation for various body types. **Results:** For individuals with a WHR of 0.85, a 35° angle provided superior image quality, while 45° was more effective for slimmer patients. This optimized approach indicates the cost-effectiveness and diagnostic value of traditional X-ray imaging. **Conclusions:** The 35° angulation standardizes lumbar X-ray imaging for an Asian cohort, reducing repeat scans and improving accuracy. Using a positioning device further enhances image quality and patient comfort, supporting the clinical utility of traditional radiography in resource-limited environments.

## 1. Introduction

Spondylolysis is a fracture or defect of the bone between the joints, mainly due to long-term injury, and is the leading cause of back pain in children and adolescents [1,2,3,4,5,6]. Commonly observed in young weightlifters and gymnasts, this stress fracture affects the pars interarticularis of the lumbar vertebrae and is often triggered by activities such as skiing, falling from a low place, or repetitive injuries [7,8,9]. Clinical presentation typically involves the lower lumbar spine and may manifest as asymptomatic or lead to spinal instability, back pain, or radiculopathy. In severe cases, it can progress from spondylosis to spondylolisthesis, often accompanied by symptoms of nerve root compression.

Clinically, lumbar spine radiographs are commonly employed to assess various conditions, including lumbar degenerative diseases, intervertebral disc stenosis, intervertebral disc herniation, spondylolisthesis, and compression fractures [10,11,12]. The common diagnostic method is to evaluate lumbar spine images using anteroposterior and lateral views [13,14]. Criteria for acceptable radiographic examinations include the absence of scoliosis and vertebral body displacement, normal physiological curvature, no narrowing of the intervertebral space, and complete pyramidal bone visibility. If the lateral image is unclear, an oblique view may be added to enhance the visibility of the upper and lower articular processes, transverse processes, isthmus, and pedicles. However, an oblique view may not present the defect clearly because the patient is not positioned with the correct posture. Hence, some studies have emphasized that lateral oblique radiographs should not be considered a definitive investigation for spondylolysis [15,16]. Therefore, many advanced countries and medical institutions choose computed tomography (CT) or single photon emission computed tomography as the gold standard of diagnosis instead of traditional radiographic examination.

Many radiologists suggest that magnetic resonance imaging (MRI) can replace CT [17,18,19,20,21]. However, four-view lumbar spine examinations remain the first step in many medical institutions in developing countries worldwide. Therefore, the four-view lumbar examination still has clinical value [22,23]. To maintain high-quality images that enable accurate diagnosis and appropriate treatment, radiologists must ensure that patient posture and imaging conditions align with differential diagnostic criteria during positioning photography [24,25]. This practice helps reduce the likelihood of repeated X-ray exposure. Optimizing radiation doses is crucial to minimizing the risk of potential medical malpractice. However, textbooks recommend an angle range for this examination of 30° to 50°, which makes standardization challenging owing to technician skill variations [26]. Moreover, Asian groups have lower lumbar lordosis, potentially making these imaging angles unsuitable for them [27,28,29]. When arranging for imaging, technicians typically instruct patients to lie flat and then turn sideways or stand upright and rotate the body in two ways. However, lying down and twisting can induce further pain in patients with lower back lesions. The abovementioned cases of lumbar disease are often associated with pain, varied body shapes (fat or thin), and challenges in achieving the most suitable angle. Maintaining a flat surface at the shoulder/hip can lead to technician misjudgments, impacting the positioning angle and overall radiographic angulation quality.

In the initial phase of this study, an investigation was conducted on whether different body shapes affected the inspection angle. Therefore, incorporating body mass index (BMI) and waist-to-hip ratio (WHR) as variables was proposed, along with different camera angles (35, 45, and 55°). The discussion focused on determining whether adjustments for different body shapes were statistically warranted. In the future, technicians could efficiently adapt the inspection angle based on the patient’s BMI, aiming to minimize errors caused by improper angles and, consequently, reduce the occurrence of bad film and the need for re-illumination.

In the subsequent phase of our research, 3D cutting technology was employed to develop a custom device that helped maintain correct posture during photography. Three-dimensional printing, considered a key component of the third industrial revolution [30], involves layer-by-layer manufacturing using a digital model created using computer-aided design (CAD) software [31,32,33]. This technology offers several advantages over traditional manufacturing methods, including high structural design freedom, low production costs in small batches, and rapid prototyping of customized products. Three-dimensional cutting, a detailed cutting process using CAD, complements 3D printing and is particularly relevant in the biomedical field [34,35]. This technology allows the precise cutting of molding materials tailored to individual rapid prototyping manufacturing characteristics. Building on the findings of the initial experiment, it was planned to use these data to determine the optimal angle and incorporate the contour parameters of various standard body shapes. Subsequently, by employing 3D cutting, a body-shaped contour assisting device that can be customized based on individual characteristics was created. When used by a patient, this device can reduce the distortion between the shoulders and hips, enhance the accuracy of photography, and ensure greater patient comfort during examinations.

## 2. Materials and Methods

### 2.1. Case Collection Method

In this study, a regional hospital was selected to investigate the correlation between the lumbar spine oblique examination angle and BMI/WHR. A questionnaire was designed to collect basic information about the participants. The questionnaire included questions regarding sex, age, BMI, and WHR.

This research commenced on 20 June 2023, following review and approval by the Institutional Review Board (IRB No. B202305043). The project host provided an explanation of the pertinent sampling methods, rights, obligations, sample collection, and relevant information to candidates in the outpatient department. All patients provided informed consent. A total of 100 walking cases were included in this study, including 47 males and 53 females.

### 2.2. Radiological Preparation

A radiography machine (Arcoma AB model 0180 SC, Arcoma AB, Växjö, Sweden) with a vertical bucky stand was used to estimate the relationship between BM, WHR, and imaging angle. With a standardized shooting distance of 100 cm (40 in), automatic exposure control was activated, and the controlled radiation dose was set at 100 kV, 200 mA, and 125 mS.

Three footboards were designed, each marked with footprints labeled 35, 45, and 55°, respectively (Figure 1).

Thus, candidates could easily adjust the tilt angle according to the markings, and researchers could capture images after making minor adjustments. Before the examination, the principal investigator explained the process, signed the patient consent form, measured the height and weight to obtain the BMI, and assessed the waist/hip circumference. The same radiographer guided the examinee to stand on different footboards throughout the examination. Based on the marked angles, both left and right shots were captured once, resulting in six groups of images (Figure 2).

These images were marked according to the imaging angle and direction (Figure 3).

To assess image quality, a scoring scale for radiologists was devised encompassing the number of visible lumbar vertebral bodies, structural integrity, sharpness, and exposure of the upper and lower facet joints, transverse process, isthmus, pars interarticularis, and pedicle (Scottie dog). The Likert scale was used for scoring statistics. Radiologists from three hospitals were invited to participate in this study. Three sets of scale lists were collected and the scoring data aggregated to form the basis for statistical analysis.

### 2.3. Statistical Analyses

Statistical data were obtained for 100 patients. Each case contained three angles, with six images on the left and right sides. Three groups of radiologists from different hospitals provided ratings, with 300 groups of reports. The BMI and WHR were included as variables. Fisher’s exact test was used to verify whether there was a correlation or significant difference in the values between groups.

## 3. Results

In this study, data from 100 candidates were collected using three sets of imaging data at different angles using a custom-built footboard, with each comprising two images from the right and left sides. Subsequently, the scores from three different sets of physician ratings were combined and included patient BMI and WHR as statistical variables. In total, 300 list scores were obtained. The male-to-female ratio was 47:53, and the median WHR was 0.85 (Table 1).

Analysis of physician ratings was a consensus among the three groups when comparing images obtained at 35, 45, and 55° angles. The results consistently favored images captured at a 35° angle over those obtained at the other two angles (Table 2). The questionnaire also sought feedback from each rating physician on the appropriateness of the shooting angle, with the results indicating that a 35° shooting angle was the most suitable.

Nevertheless, there were discrepancies in the results for the left and right sides. Despite a unanimous preference for 35° as the most suitable angle among all three groups of doctors, there was greater consistency on the left than on the right. The imaging scoring data with BMI and WHR were included in the statistical system for comparison, and the resolution, clarity, and completeness of the images in each group were not significantly related to BMI (Figure 4).

For a median WHR of 0.85, an angle of 35° was deemed most suitable; however, as the WHR increased to 0.90, the optimal angle increased to 45° (Figure 5). Thus, the image quality associated with the shooting angle and WHR was found.

In this study, a 35° imaging angle was determined to be optimal for individuals with an average body shape. However, 45° also demonstrated advantages for extremely slim body types, although the results did not show any statistically significant differences.

The findings from this study in collaboration with 3D printing companies were employed. Using the average Asian human back profile formula from a comprehensive database, the author manufactured an EVA pad as an assisted device by 3D cutting technology. This adaptive board was combined with a 35-degree firm foam pad as a positioning aid. Re-examination was conducted to assess this assistive device’s impact on artifacts and X-ray resolution. Following confirmation by the radiologist that there was no apparent impact, the auxiliary device was used. Table 1 illustrates the characteristics of the candidates in this study. Overall, 100 candidates were recruited (male-to-female ratio, 47:53). The median BMI was 23.16, and the median WHR was 0.84. The candidates were relatively healthy. No previous surgery for the spine or lower extremities was recorded. Table 2 demonstrates the optimum shooting angle.

Three groups of doctors considered 35° as the optimal angle. However, there were variations between the right and left sides. In the right-side group, three doctors partially agreed that 35° was sufficient. The Fisher’s exact test *p*-value is 0.0009, which is statistically significant. The *p*-value on the left side is 0.1247. Regarding evaluating the most appropriate angle, the data on the left side has no obvious relationship with the physician’s interpretation habits.

In this study, the relationship of influence between the optimum angle of 35° and 45° by BMI and WHR was analyzed. The author figured out that the influence of WHR is larger than BMI, especially on the left side. However, the right-side data shows no obvious relationship between the optimum angle and WHR and BMI.

## 4. Discussion

Pars fracture is a common cause of chronic pain in adolescents, which also affects middle-aged adults. Historically, neurosurgeons relied on oblique lumbar spine examination to identify vertebral fractures. Owing to its poor resolution, advanced countries in Europe and the United States instead employ CT and MRI. However, not every country or medical institution can access MRI for routine testing. The principal weakness of the traditional lumbar oblique examination was found to be its loose threshold. The loose restriction of imaging angles from 30° to 45° is easily affected by the subjective habits of radiology technicians. Both CT and MRI offer significant advantages in identifying pars fractures, providing high accuracy and reconstructable multi-angular sectional images that enable doctors to assess damage from different perspectives. However, the disadvantages of these technologies relative to simple X-ray imaging are that CT involves a higher radiation dose, MRI requires a longer time to perform, and both are relatively expensive. By contrast, the advantages of traditional oblique examinations include low radiation doses and low health insurance costs. However, their disadvantages include long positioning time and low accuracy. Therefore, designing an assistive device that fits contours and that could solve this problem was proposed.

This study included 100 candidates to investigate the relationship between BMI, WHR, and shooting angle. According to the statistical analysis results, the best shooting angle was 35°. However, this result had no significant relationship with BMI. When the median WHR was 0.84, doctors unanimously deemed 35° the most suitable angle.

The authors concur with the notion that BMI may not highlight the difference in body shape as effectively as WHR [36,37,38]. Data from the three groups of radiologists showed that this angle was reasonable and correct. At this angle, the maximum number of vertebrae can be resolved. The resolution and clarity were better than those at the other angles. However, when analyzing the results, the scores on the left and right sides were significantly different. The report for the left side found that the scoring physicians had a high degree of agreement at 35°, indicating that there was no influence of physician subjectivity on this part. However, there was a disagreement regarding the appropriateness of the angle in the report for the right side. While WHR and BMI did not have an obvious influence on this angle, the influence of WHR on the left side was slightly more significant than that of BMI. As the three groups of doctors did not know each other, identifying the reasons for such differences would require more in-depth exploration. In addition, for very slender individuals, it was found that a 45° shooting angle could provide clear images. Although there was no statistically significant difference in this result, 45° is probably an option when arranging examinations for very slender patients in the future. Based on the results of this study, the proper shooting angle for oblique lumbar spine examinations can be redefined to be 35°. A footboard was designed to ensure that the patient could maintain the correct angle for the examination. Additionally, an assistive device for patients who cannot walk was manufactured (Figure 6), enabling radiology technicians to easily position patients.

### 4.1. Limitations

Lumbar oblique X-ray imaging remains a valuable tool for diagnosing lumbar spondylolysis, particularly in resource-limited settings; however, their use has declined in advanced hospitals where CT and MRI provide more comprehensive imaging options. Consequently, there is a relative lack of recent studies on lumbar oblique X-ray imaging, which presents a challenge in contextualizing our findings with contemporary data. Additionally, during this study, cases of congenital scoliosis were encountered, which impacted the quality and consistency of oblique imaging results. While the inclusion of these cases could introduce variability, these were included in the analysis, considering the relatively high incidence of scoliosis among young individuals in Taiwan and its frequent occurrence at mild curvature angles (28–30°). This inclusion provides a more representative analysis of populations with similar demographic characteristics.

### 4.2. Future Attributions

In many developed countries, neurosurgeons and orthopedic surgeons increasingly favor CT over lumbar oblique X-ray imaging owing to CT’s superior image quality and diagnostic power. However, in developing regions, economic limitations often make CT and MRI impractical, resulting in continued reliance on oblique X-ray imaging as an accessible diagnostic option. Our research addresses this need by proposing an optimized imaging angle and developing a 3D-printed positioning device to standardize patient posture, enhance imaging consistency, and reduce technician adjustment time. This approach not only simplifies positioning but also minimizes patient discomfort, providing a practical solution for facilities with limited access to advanced imaging.

This study highlights the ongoing relevance and potential of lumbar oblique radiography in specific clinical contexts, encouraging a re-evaluation of their diagnostic applications. By refining radiographic techniques and incorporating positioning aids, the authors aim to enhance the reliability and utility of oblique imaging, especially where advanced modalities are not readily available. Researchers should expand on these findings by conducting future studies across diverse populations and healthcare settings to assess the adaptability of these techniques to different demographic and body type variations. Larger-scale studies with broader population samples are essential to validate the efficacy and generalizability of the proposed methods, ultimately helping to establish guidelines that can be applied across varied clinical and regional contexts.

## 5. Conclusions

Spondylolisthesis is a prevalent cause of back pain across various age groups, and its diagnosis often relies on radiographic assessments, with lumbar spine X-ray imaging remaining a common initial approach in many clinical settings. However, traditional oblique X-ray imaging frequently yields suboptimal images due to challenges in maintaining consistent posture. This study seeks to enhance the precision and efficiency of oblique X-ray imaging while minimizing patient discomfort. By testing various imaging angles and factoring in BMI and WHR, a 35° angle was identified as optimal for achieving high-quality images in our cohort.

To support this angle, a specialized footrest and triangular auxiliary pad were designed and paired with a 3D-printed back panel tailored for the average Asian body profile. This customized device stabilizes the shoulder-to-waist axis, ensuring accurate posture alignment and enhancing image clarity. These improvements can decrease exam duration, reduce repeat imaging needs, and increase equipment usage rates. Additionally, this approach offers notable cost and scheduling benefits, making it a practical solution even in settings where CT and MRI are accessible. This technique, especially in resource-limited environments, may serve as an efficient and cost-effective enhancement to traditional X-ray imaging for lumbar disorders.

## Figures and Tables

**Figure 1 jfmk-10-00023-f001:**
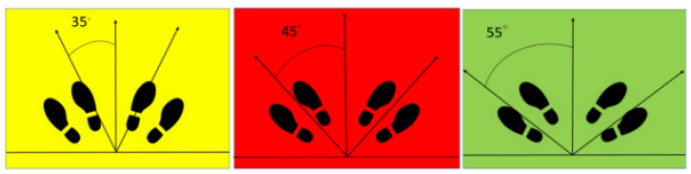
Three different angle markings of the footboard are shown. These boards are color-coded to prevent technicians from being confused. Positioned in front of the vertical bucky stand, candidates can easily locate their standing position based on the footprints.

**Figure 2 jfmk-10-00023-f002:**
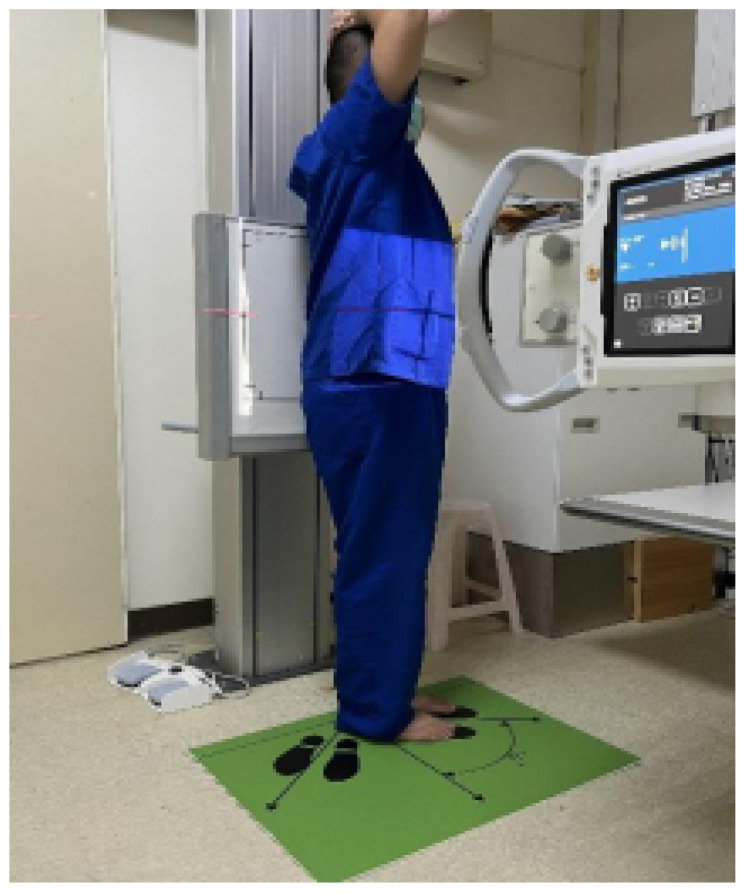
Use of the footboard. This figure illustrates a candidate standing on the footboard and respecting the footprint angle easily. Minor adjustments are conducted following the arrow. This way, the researcher can obtain an oblique examination under the correct shooting angle.

**Figure 3 jfmk-10-00023-f003:**
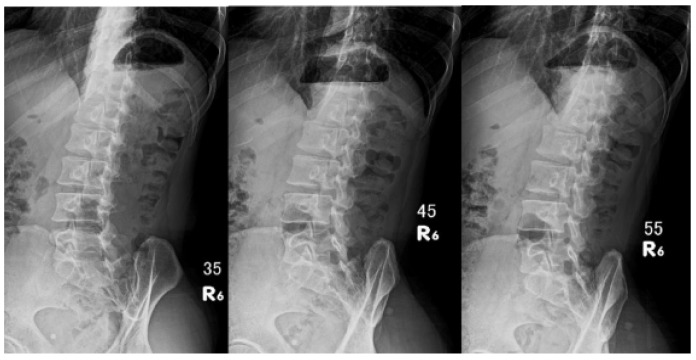
Image from three different angles. The technician marked the actual shooting angle and direction on the film. Thus, the radiologist can clearly distinguish each image. Image files are recorded according to case numbers to protect patient privacy.

**Figure 4 jfmk-10-00023-f004:**
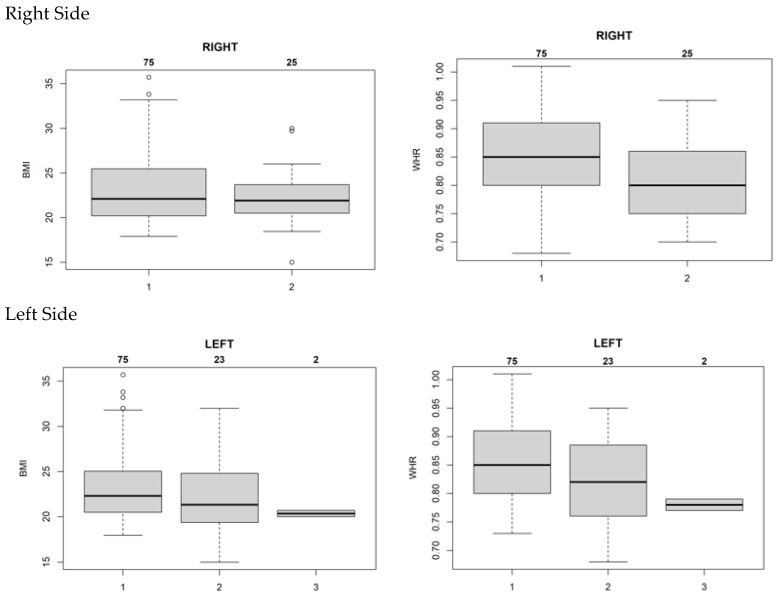
Scoring consistency between optimum angle and median WHR. This figure illustrates the sufficiency of the imaging angle compared with BMI and WHR. The number 1 means the ratings are all consistent, 2 means two people are consistent, and 3 means none. When the median WHR is 0.85, the consistency among the three doctors is the best. Meanwhile, BMI did not show an obvious trend to support this consistency.

**Figure 5 jfmk-10-00023-f005:**
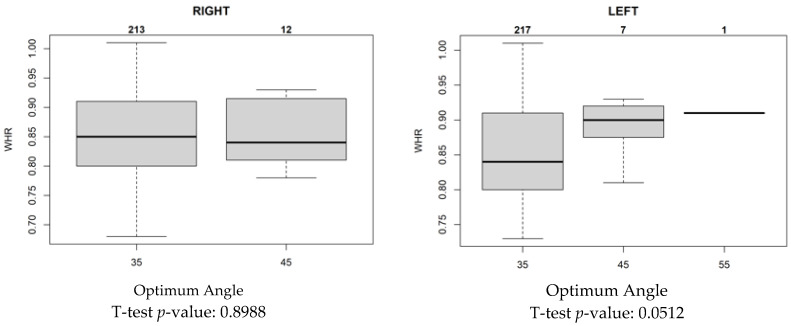
Relationship between optimum angle and median WHR. This figure points out that for a median WHR of 0.85, 35° was deemed the most suitable angle; however, as the WHR increased to 0.90, the optimal angle increased to 45°. Moreover, there is a large difference in WHR between the two groups at 35° and 45°, but no significant difference in BMI.

**Figure 6 jfmk-10-00023-f006:**
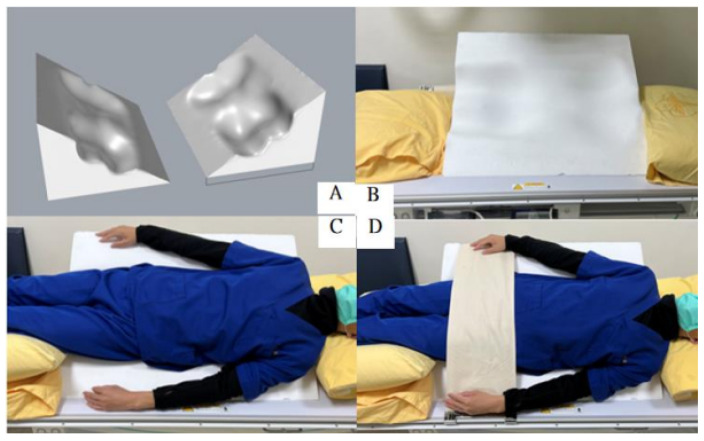
Three-dimensional (3D)-assisted device in action. This figure shows the 3D-assisted device designed and used by the authors. (**A**) shows a 3D design drawing of our assistive device. (**B**) illustrates the device placed on the examination table. (**C**) is a schematic diagram of a medium-sized male imitating an individual placed on an assistive device. Shoulders and hips remain equally aligned. However, pillows needed to be placed on the head and lower limbs to maintain the examination posture. In subsequent modifications, the length of the assistive device can be increased to increase comfort. In (**D**), the restraint straps attached to the examination equipment are used to assist in positioning and make the inspected patient feel more secure.

**Table 1 jfmk-10-00023-t001:** Participant characteristics.

Variable	Included in the Analysis
Gender (Male/Female)	47:53
BMI (Kg/m^2^)	23.16 * (±4.09 **) IQR: 20.25–25.00
WHR	0.84 * (±0.08 **)

* Mean, ** Standard deviation.

**Table 2 jfmk-10-00023-t002:** Physician’s Rating of Photography Angle.

	Right 35°	Right 45°	Right 55°	Left 35°	Left 45°	Left 45°
Doctor A	100	0	0	100	0	0
Doctor B	95	5	0	95	4	1
Doctor C	89	11	0	96	4	0

Fisher’s Exact Test *p*-values are 0.0009 * (right side) and 0.1247 (left side). * Statistically significant value.

## Data Availability

The datasets used and analyzed during the current study are available from the first author upon reasonable request due to privacy and ethical restrictions.

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
