# Peer review of "Evaluation of Optimized Lumbar Oblique X-Ray Angles with Positioning Assistance for Enhanced Imaging Quality: A Pilot Study in an Asian Cohort"

_jfmk, 2025, doi:10.3390/jfmk10010023_

Round 1

Reviewer 1 Report

Comments and Suggestions for Authors

Review of “Evaluation of Optimized Lumbar Oblique X-Ray Angles with Positioning Assistance for Enhanced Imaging Quality: A Pilot Study in an Asian Cohort” by Yu-Li Wang, Hsin-Yueeh Su, Chao-Min Cheng and Kuei-Chen Lee

The study examines the optimal angles for lumbar oblique X-rays in an Asian population. It identifies a 35° angle as the most suitable for most patients, considering waist-to-hip ratio (WHR) and body mass index (BMI). For very slim individuals, a 45° angle might be more appropriate. Using a customized positioning footboard, the study shows significant improvements in image quality, reducing the need for repeat scans and increasing patient comfort.

This work represents a relevant contribution to medical imaging, particularly in resource-limited clinical contexts. It will likely be well-received by the readership of Journal of Functional Morphology and Kinesiology.

The reviewer has the following (minor) remarks:

  1. The manuscript cites only 9 references dated after 2010 out of a total of 30, and only 2 after 2020. It is recommended to include more recent studies to better contextualize the limitations of the current standard X-ray angles (30°-50°) and to align the discussion with contemporary advancements in the field of medical imaging. Adding up-to-date references will strengthen the relevance and impact of the study.
  2. For a more formal and neutral academic tone, the use of "we" should be avoided. The authors may rephrase the text in a passive voice.
  3. Figures 4 and 5 appear compressed and lack clarity.

Reviewer 2 Report

Comments and Suggestions for Authors

Title: Evaluation of Optimized Lumbar Oblique X-Ray Angles with Positioning Assistance for Enhanced Imaging Quality: A Pilot Study in an Asian Cohort

Outline: This study assessed whether the standard radiographic angles suggested in textbooks are optimal for an Asian population, finding 35° to be the most effective angle for lumbar oblique X-ray imaging. From this study, the authors concluded that the 35° angulation standardizes lumbar X-ray imaging for an Asian cohort, reducing repeat scans and improving accuracy. Using a positioning device further enhances image quality and patient comfort, supporting the clinical utility of traditional radiography in resource-limited environments.

Critique:

1. I think this study is very interesting, and well-written. Thus, I think current version is also deserves for acceptance in this journal. However, I have several minor comments in this study.

2. In introduction, please describe characteristics of asian population in lumbar spine, presenting low lumbar lordosis than in European population et al.

3. Were the selected participants voluntarily recruited, or were they part of an asymptomatic normal population?

4. Figure 4: it is difficult to see, please re-upload for easy-to-see.
